# Tuning porosity in macroscopic monolithic metal-organic frameworks for exceptional natural gas storage

B.M. Connolly [1,2], M. Aragones-Anglada[2], J. Gandara-Loe[3], N.A. Danaf[4], D.C. Lamb[4], J.P. Mehta[1,2], D. Vulpe[2], S. Wuttke [4,5], J. Silvestre-Albero [3], P.Z. Moghadam [6], A.E.H. Wheatley[1] & D. Fairen-Jimenez [2]

Widespread access to greener energy is required in order to mitigate the effects of climate change. A significant barrier to cleaner natural gas usage lies in the safety/efficiency limitations of storage technology. Despite highly porous metal-organic frameworks (MOFs) demonstrating record-breaking gas-storage capacities, their conventionally powdered morphology renders them non-viable. Traditional powder shaping utilising high pressure or chemical binders collapses porosity or creates low-density structures with reduced volumetric adsorption capacity. Here, we report the engineering of one of the most stable MOFs, Zr-UiO-66, without applying pressure or binders. The process yields centimetre-sized monoliths, displaying high microporosity and bulk density. We report the inclusion of variable, narrow mesopore volumes to the monoliths' macrostructure and use this to optimise the pore-size distribution for gas uptake. The optimised mixed meso/microporous monoliths demonstrate Type II adsorption isotherms to achieve benchmark volumetric working capacities for methane and carbon dioxide. This represents a critical advance in the design of air-stable, conformed MOFs for commercial gas storage.

[1] Department of Chemistry, University of Cambridge, Lensfield Road, Cambridge CB2 1EW, UK. [2] Adsorption & Advanced Materials (AAM) Laboratory, Department of Chemical Engineering & Biotechnology, University of Cambridge, Philippa Fawcett Dr, Cambridge CB3 0AS, UK. [3] Laboratorio de Materiales Avanzados, Departamento de Química Inorgánica-Instituto Universitario de Materiales, Universidad de Alicante, Ctra. San Vicente-Alicante s/n, E-03690 San Vicente del Raspeig, Spain. [4] Department of Chemistry, Center for NanoScience (CeNS), Nanosystems Initiative Munich, Center for Integrated Protein Science Munich, Ludwig-Maximilians-Universität, München (LMU), Butenandtstrasse 11, 81377 Munich, Germany. [5] School of Chemistry, College of Science, University of Lincoln, Brayford Pool, Lincoln LN6 7TS, UK. [6] Department of Chemical and Biological Engineering, University of Sheffield, Mappin Street, Sheffield S1 3JD, UK. Correspondence and requests for materials should be addressed to A.E.H.W. (email: aehw2@cam.ac.uk) or to D.F.-J. (email: df334@cam.ac.uk)

With global energy usage predicted to exponentially increase over the next 50 years, widespread access to clean fuels is imperative[1]. Natural gas (NG), mainly composed of methane, is a favourable fuel due to its abundant natural reserves and reduced pollutant emissions relative to other fossil fuels[2]. Although liquefied NG at 113 K and compressed NG (CNG) at 250 bar are already in use, cryogenic and high-pressure storage methods have severely limited practical applications[3]. For example, the higher pressures required for liquification can only be achieved using complex and expensive multi-stage compressors. To counteract these limitations, the U.S. Department of Energy (DOE) sets the ambitious volumetric NG storage target of 263 $cm^3$ (STP) $cm^{-3}$ at 65 bar, which is equal to that of CNG at 250 bar and 298 K[4]. The moderate 65 bar pressure targeted by the DOE is highly practical, being achieved using inexpensive compressors and light-storage tanks[5]. This offers not only safety benefits but also improvements in large-scale industrial economics. Adsorbed NG (ANG) stored at relatively low pressure on porous materials has emerged as a safe and cost-effective method of achieving this ambitious target[6]. Furthermore, advances in gas-storage technology are critical for multiple energy applications, including pollutant capture, such as $CO_2$[7]. Technological advances in adsorption are clearly vital for environmental fuel usage, both in this century and the future economy[8].

Over the last 20 years, metal–organic frameworks (MOFs)[9], obtained through the crystalline self-assembly of metal atoms/ clusters and organic molecules, have massively extended the use of porous materials in adsorption applications[10,11]. Indeed, they show promise for both dense NG storage[12,13] and selective $CO_2$ capture[14,15]. Microporous MOFs, with pore sizes below 2 nm, can exhibit outstanding total volumetric $CH_4$ uptake, reported to exceed 200 $cm^3$ (STP) $cm^{-3}$ at the DOE target pressure of 65 bar (e.g., NU-111, PCN-14 and HKUST-1)[16]. Meanwhile, an extraordinary $CO_2$ capacity of 320 $cm^3$ (STP) $cm^{-3}$ (35 bar) was calculated for MOF-177[17].

However, despite the immense potential of MOFs evident from these benchmark results, each of these quoted volumetric uptakes are only theoretical storage maxima that do not take into account the working capacity or the real density of the material. Figure 1 illustrates these ideas. Firstly, the working capacity, in contrast with the total uptake, represents the true volume of gas available under practical working conditions. Volumetric working capacity is defined as the volume adsorbed at the maximum working pressure minus the uptake at the minimum desorption pressure (typically ca. 5 bar). Thus, the Type I isotherm shape of typical microporous MOFs leads to a low working capacity due to high uptake at low pressure and plateauing at high pressure (Fig. 1a). An example is HKUST-1, which shows an outstanding, theoretical, total volumetric capacity of 270 $cm^3$ (STP) $cm^{-3}$ (65 bar) using single-crystal density, but exhibits an accessible working capacity of 190 $cm^3$ (STP) $cm^{-3}$, measured between 5 and 65 bar[16]. Type II and IV isotherms do not plateau at high pressures, but still show a relatively high uptake at low pressure (Fig. 1b). For this reason, adsorbents that maximise working capacity via Type V S-shaped isotherms, e.g., flexible but also rigid MOFs with unusual adsorption behaviour[18], are of significant interest (Fig. 1c). For example, Long et al. reported an outstanding theoretical 197 $cm^3$ (STP) $cm^{-3}$ volumetric working capacity in the flexible MOF Co(bpd) and a theoretical volumetric capacity of 203 $cm^3$ (STP) $cm^{-3}$, both calculated using the single-crystal density of the MOF[19]. Another example is the dynamic copper-based MOF UTSA-76a reported by Chen et al., with a benchmark theoretical working capacity of ca. 200 $cm^3$ (STP) $cm^{-3}$ and a total volumetric capacity of 257 $cm^3$ (STP) $cm^{-3}$[20]. However, even these fall short of the DOE target[19].

In addition to the shape of the isotherms, although these exceptional, theoretical, volumetric capacities were determined experimentally via the measurement of gas isotherms, it is common practice to convert the obtained gravimetric uptake ($cm^3$ $g^{-1}$) to volumetric capacity ($cm^3$ $cm^{-3}$) using the MOFs' ideal single-crystal density. This neglects the detrimental effect of extensive interstitial space that exists between the synthesised MOF particles in the powder (Fig. 1d). Experimentally, the bulk or tap density of such finely powdered materials is very low, and this loose packing of the MOF particles is widely observed to reduce the adsorbents volumetric capacity. Although a 25% reduction in packing efficiency is widely reported[21], experimentally compacted MOF pellets have even been reported with a bulk density 50% lower than the theoretical crystal density[16]. Taking into account the real density of the powder, the reported benchmark working capacities of flexible MOFs can be expected to fall between 100 and 150 $cm^3$ (STP) $cm^{-3}$. With this loss of storage capacity in mind, the volumetric DOE target for unpacked and undensified NG adsorbents becomes a staggering 350 $cm^3$ (STP) $cm^{-3}$[22], making it clear that, in terms of practical volumetric working capacity, powdered MOFs have not yet achieved industrially viable NG storage capacities.

Attempts to overcome the low densities of MOF powders have led to densification using pressure, with chemical binders added to improve the mechanical properties of the shaped pellets[23,24]. However, the application of high pressures to MOFs provokes devastating and irreversible collapse of the internal porosity, while binders might block pores and the use of milder pressures prevent efficient densification. The result of these traditional pelletization procedures has therefore been low-density pellets with diminished volumetric capacity. Furthermore, many MOFs are incompatible with this pelletization process. For example, flexible Co(bpd) undergoes a staggering 94% increase in unit cell-volume upon transition from its collapsed to expanded state during $CH_4$ adsorption. In addition to issues with the mechanical properties of flexible MOF pellets, they would require highly voluminous tanks to allow the expansion of these constrained volumes. Long et al. reported that, upon densification of Co(bpd) powder to only 68% of the maximum crystal density, 11% of collapsed Co(bpd) crystals were no longer able to expand in the constrained volume[19]. Clearly, the commercialisation of gas-storing MOFs has been hugely limited by the non-optimised volumetric capacities in the conformed materials[25,26].

Much effort has been applied to finding alternative ways to pelletise, densify and shape MOFs. As a significant advancement in this direction, we reported the sol–gel synthesis of centimetre-scale monolithic MOFs (monoMOFs) monoZIF-8[27] and mono-HKUST-1[28] without using supports, applied pressures or binders. In particular, the high microporosity and bulk density of mono-HKUST-1 enabled a record total $CH_4$ storage capacity of 259 $cm^3$ (STP) $cm^{-3}$, making it the first conformed material to reach the U.S. DOE target. Although this material has laid the groundwork for MOF densification with retention of physical properties, from an industrial perspective, it failed to address significant practical barriers. Again, its high microporosity and a strong physical interaction with NG at low pressure leads to a working capacity that falls short of the DOE target. As a result of the classical Type I isotherm this MOF exhibits (Fig. 1a), moderately increasing the maximum pressure to a still industrially viable 100 bar[29] fails to achieve the working capacity target of the Advanced Research Projects Agency-Energy (ARPA-E). This is a persistent characteristic of purely microporous MOFs; micropores rapidly saturate at low pressure with only marginal gains achieved by applying higher pressures. Finally, the high water affinity of the metal nodes in HKUST-1 incurs irreversible loss of crystallinity and adsorption performance under moisture in ambient

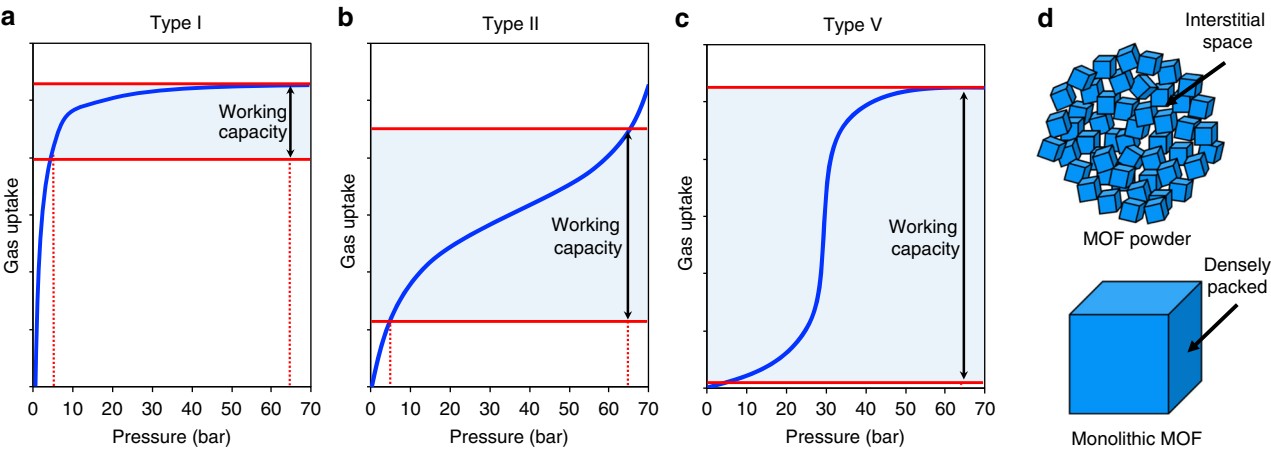

**Fig. 1** Gas adsorption isotherms and particle densification. **a–c** Variations in gas adsorption isotherms and corresponding changes in working capacity for Type I, II and V isotherms, respectively. **d** Representation of the abundant interstitial space amongst non-densified MOF particles compared with a densified monolith

conditions[30,31]. This makes long-term usage of the adsorbent problematic. It is the inadequate working capacity and poor chemical stability of even benchmark monoHKUST-1 that has driven us to further develop a synthetic procedure for enhancing the volumetric working capacity of high-stability MOFs with tuneable mesoporosity.

Among all the MOFs, UiO-66 ($Zr_6O_4(OH)_4(1,4$-benzenedicarboxylate)$_6$ (Fig. 2a) represents the prototypical structure for high porosity and thermo-chemical/mechanical stability[32]. This high-stability MOF has typically been overlooked as a NG adsorbent as a result of its exclusive microporosity and relatively poor physical interaction with $CH_4$—especially when compared with MOFs with more exotic surface chemistry. However, in our view, this weak interaction at low pressure can be utilised to enhance its working capacity. Through the development of a synthetic protocol that introduces tuneable volumes of mesoporosity into the monolith macrostructure, we modify and optimise pore-size distribution to convert this MOFs undesirable Type I adsorption isotherm into a promising Type II isotherm (Fig. 1b). The result is a mixed micro-/mesoporous monolith with a volumetric working capacity that surpasses the theoretical maximum for the traditional, microporous MOF. Using grand canonical Monte Carlo (GCMC) simulations, we are then able to rationalise our observations and to account for the outstanding values recorded for volumetric NG and $CO_2$ adsorption.

## Results

**Synthesis of monolithic MOFs**. In our previous syntheses of monoZIF-8 and monoHKUST-1[27,28], MOF primary nanoparticles (70 and 50 nm, respectively) were densified during drying to yield centimetre-scale monoliths. We demonstrated that both mild drying conditions using a low-surface tension solvent and small primary particles were critical to monolith formation. For monoUiO-66, we have modified a literature synthesis for UiO-66 gel[33] to produce 10 nm primary MOF particles in a viscous gel phase (Fig. 2c). We then washed and dried the MOF gel under four different conditions (Supplementary Table 1) to obtain macroscopic monoliths (Fig. 2d–g). monoUiO-66_A was prepared by the existing literature procedure[33], whereby reaction solvent (DMF) and impurities were washed from the primary particles with ethanol before drying at 200 °C. This fails to create large, conformed monoliths, giving exclusively sub-mm-sized MOF fragments (Fig. 2d).

We evaluated the role of drying temperature in monolith formation. In the case of monoZIF-8 and monoHKUST-1, high

temperatures induced fast removal of solvent from the interstitial spaces between primary particles. This stress at the vapour–liquid meniscus interface destroyed the materials' gel macrostructure, resulting in standard powders. We decided therefore to synthesise monoUiO-66_B by again washing the gel in ethanol, but instead drying at 30 °C. This resulted in centimetre-sized, opaque monoliths (Fig. 2e), supporting the view that the drying temperature is fundamental to controlling the monolithic macrostructure. We further studied the influence of washing solvent on monolith physical properties by next washing the DMF synthesised gel in further DMF before drying at 30 °C; this produced optically transparent monoUiO-66_C (Fig. 2f). This solvent-induced alteration in transparency demonstrates the complex range of solvent–particle interactions that exist. We have previously proposed that slow evaporation of the reaction solvent, coupled with the presence of residual precursors, facilitates primary particle interaction by effectively extending the reaction time[28]. We propose that epitaxial growth of the primary particles reduces the electron density contrast and therefore reduces the optical visibility of the interparticle barrier, giving a transparent material[34]. According to this view, the use of ethanol to wash UiO-66 particles quenches the reaction, whereas the use of DMF facilitates its continuation. Furthermore, while the surface tension of DMF is higher than that of ethanol, the boiling point of DMF (153 °C) is also significantly higher (78 °C), meaning that the evaporation of DMF from the primary particles is slower. Given the high mechanical strength of UiO-66, the balance between surface tension and drying speed permits the gel macrostructure to be maintained throughout the drying process via gradual solvent evaporation. Finally, we used a similar synthesis for monoUiO-66_D (Fig. 2g), but with an extended (180 min) centrifugation step. This was done to better understand the effects of primary particle densification prior to drying. These relatively minor changes to the synthetic procedure incurred fascinating changes in the physical properties of the resulting monoliths.

The powder X-ray diffraction (PXRD) patterns for each of monoUiO-66_A-D show significant Scherrer line broadening. This stems from the non-convergence of diffraction peaks in nanocrystallites (Fig. 2b)[35]. Transmission electron microscopy (TEM) of monoUiO-66_D further confirms that the monoliths comprise densified MOF nanoparticles with particle interstitial space visibly reduced compared with that in the MOF gel (Supplementary Fig. 1a–d). Low-magnification scanning electron microscopy (SEM) shows the smooth surface of the macrostructure, which

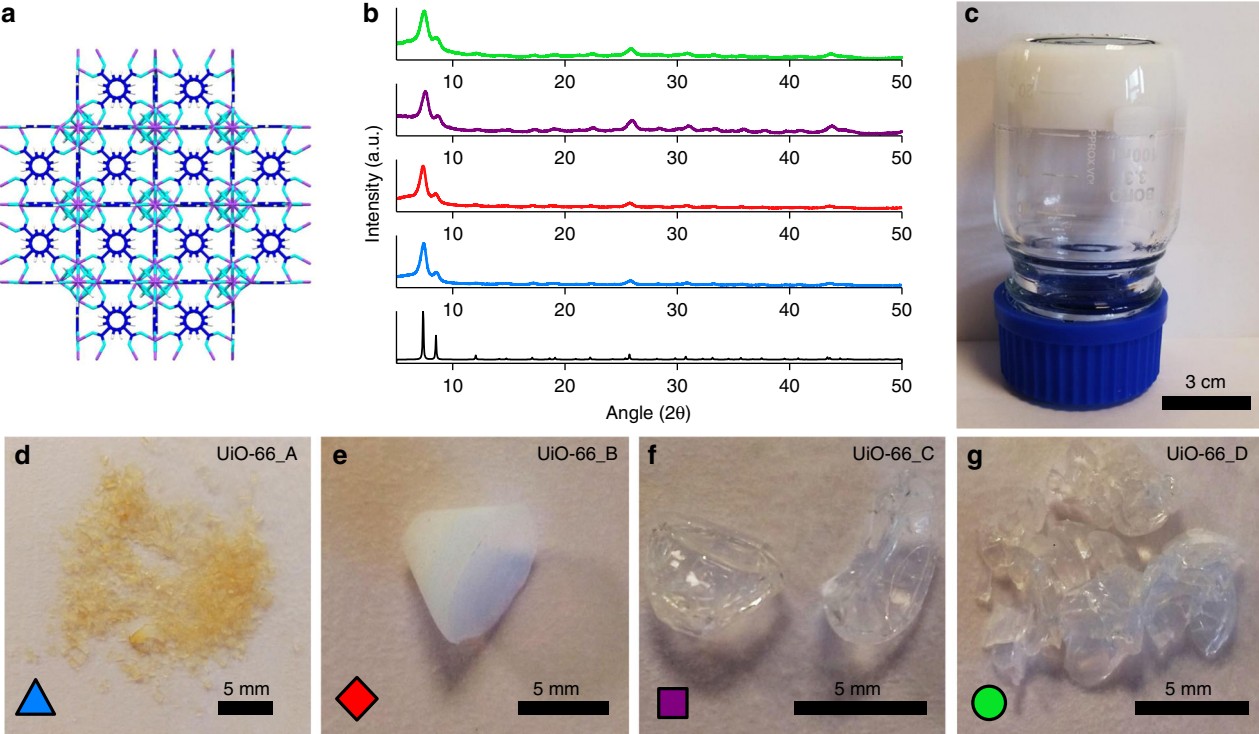

**Fig. 2** Crystalline structure and optical images of UiO-66 gel and monoliths. **a** Graphical representation of the microporous crystal structure of UiO-66 with purple, sky blue, navy blue and white sticks representing zirconium, oxygen, carbon and hydrogen atoms, respectively. **b** Comparison of simulated XRD pattern of UiO-66 generated from its crystal structure; black, with PXRD patterns of $_{mono}$UiO-66 samples ($_{mono}$UiO-66_A; blue, $_{mono}$UiO-66_B; red, $_{mono}$UiO-66_C; purple, $_{mono}$UiO-66_D; green) confirming successful synthesis in each case. **c** UiO-66 gel used for synthesising monoliths. **d** UiO-66 washed in ethanol and dried at 200 °C ($_{mono}$UiO-66_A). **e** UiO-66 washed in ethanol and dried at 30 °C ($_{mono}$UiO-66_B). **f** UiO-66 washed in DMF and dried at 30 °C ($_{mono}$UiO-66_C). **g** UiO-66 washed in DMF with extended centrifugation and then dried at 30 °C ($_{mono}$UiO-66_D). **d**–**g** (lower left corner, inset), symbols used to represent each material in Figs. 4 and 5

resolves into a homogenous array of densely packed nanoparticles at high magnification (Supplementary Fig. 1e, f). These combined data support the proposed mechanism of monolith formation via primary particle densification. Elemental analysis showed that all four monoliths have compositions closely matching the crystal structure of defect-free UiO-66 (Supplementary Table 2). The high thermal stability of the monoliths is demonstrated by thermal decomposition above 500 °C (Supplementary Fig. 2), which also matches literature values[32]. In addition, the characteristically high mechanical strength of UiO-66[36] was reproduced by the four monoliths ($E = 4.2$–$14.3$ GPa, $H = 0.11$–$0.48$ GPa) (Supplementary Fig. 3). These outstanding values are also comparable with those of the robust monoliths we have previously reported, $_{mono}$ZIF-8 ($E = 3.6 \pm 0.2$ GPa, $H = 0.43 \pm 0.03$ GPa)[27] and $_{mono}$HKUST-1 ($E = 9.3 \pm 0.3$ GPa, $H = 0.46 \pm 0.03$ GPa)[28].

To further elucidate the synthetic origins of their different textural properties, we studied $_{mono}$UiO-66 using fluorescence lifetime imaging microscopy (FLIM). FLIM utilises spatially resolved fluorescence lifetime decays to investigate the nanoscopic environment of a material[37], such as its morphology and defects[38]. FLIM reveals the samples to be comprising smaller particle aggregates with distinct fluorescence lifetimes (Fig. 3a, b). Interestingly, $_{mono}$UiO-66_A shows high homogeneity, being comprised entirely of quazispherical aggregates with uniform fluorescent lifetimes of ~4.7 ns (orange), whereas $_{mono}$UiO-66_B-D show two distinct morphologies: small, circular particles and large rod-like particles. The small, circular particles have a long autofluorescence lifetime (~4.5 ns, orange), whereas the larger aggregates show a faster fluorescent decay (~3 ns, blue). Figure 3c shows the FLIM

data analysis by the phasor approach, graphically translating fluorescence lifetime into Fourier space (Supplementary Figs. 4–7). Here, mono-exponential decays fall along an arc of radius 0.5 with long lifetime components located near the origin (0, 0) and short lifetime components near (1, 0). Multi-exponential decays comprise a weighted vector of the constituent phasors, meaning that all decay pathways in phasor space lie within the arc[39,40]. In our case, each $_{mono}$UiO-66 sample occupies phasor space within the arc, indicating multicomponent exponential decays. A single population was recorded for $_{mono}$UiO-66_A whereas plots for $_{mono}$UiO-66_B-D are heterogeneous showing at least two populations. This can be correlated with their biphasic morphology as described above and supports the proposed relationship between monolith textural properties and synthetic parameters. Since all monoliths are synthesised from the same primary particles (ca. 10 nm), any changes in photophysical properties must be caused by the way these identical particles interact with each other under different drying conditions. $_{mono}$UiO-66_B-D show a statistically greater prevalence of larger, rod-like aggregates within the monolith macrostructure than $_{mono}$UiO-66_A. To incur these differences in fluorescence lifetime, the observed aggregates cannot be an ensemble of completely discrete particles, as they would be in a powdered MOF. Instead, the primary particles of which each aggregate is comprised must be in close physical proximity and the decrease in fluorescence lifetime could be an indication of chemical interactions. These FLIM measurements thus reinforce our understanding of the monolith formation mechanism by confirming what morphology-only SEM measurements cannot: alterations to the synthetic procedure influence primary particle interaction.

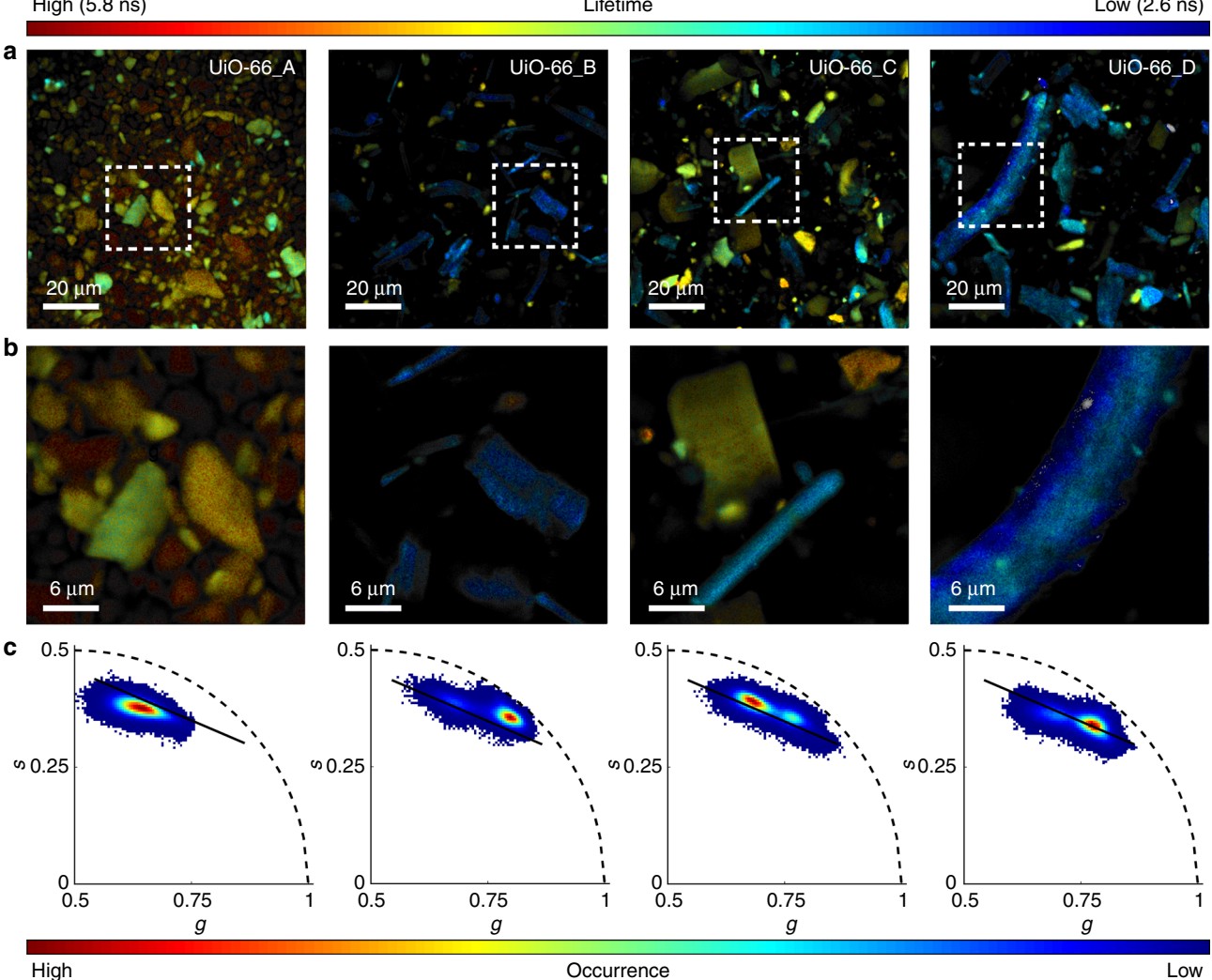

**Fig. 3** Fluorescence lifetime imaging microscopy (FLIM) studies of monoliths UiO-66_A-D. To investigate the sub-structural morphologies of the UiO-66 monoliths using FLIM, the monoliths were first gently ground into smaller pieces with a spatula such that the substructures could be monitored on the microscope. **a** Low and **b** high-magnification FLIM images of $_{mono}$UiO-66_A-D showing the aggregates that comprise each monolith. White dashed boxes indicate the area selected for high-magnification imaging. Colours correspond to excitation lifetime (see upper colour bar). **c** Two-dimensional histogram phasor plots generated from FLIM images of each monolith (an average of four images was used to generate each plot, Supplementary Figs. 4–7). Colours correspond to the frequency of occurrence (see lower colour bar)

**Adsorption properties, porosity and density of monoliths**. We analysed the porosity of the $_{mono}$MOFs through $N_2$ adsorption at 77 K. Figure 4a, b and Supplementary Fig. 8 show the adsorption and desorption isotherms, respectively; Table 1 shows the BET areas ($S_{BET}$) calculated using Rouquerol's consistency criteria (Supplementary Figs. 9–12) as well as the micro ($W_o$) and the total ($V_{tot}$) pore volumes[41]. These experimental values are lower than, but consistent with, the theoretical maximum for defect-free UiO-66 ($S_{BET} = 1644\,m^2\,g^{-1}$), calculated using GCMC simulations as well as closely matching the original report of UiO-66 by Lillerud et al. ($1187\,m^2\,g^{-1}$)[32]. For all monoliths, we observed Type IV $N_2$ isotherms, with high gas uptake below 0.1 $P/P_o$ indicating extensive microporosity. The shape of each isotherm (Fig. 4b) is characteristic of the sequential filling of the discrete tetrahedral and octahedral (8 and 11 Å diameters, respectively) cavities of UiO-66[42]. Extensive $N_2$ uptake at higher relative pressures was also recorded, indicating the existence of mesoporosity. After micropore filling, capillary condensation of the gas takes place within these wider cavities. The extent of the mesoporosity, obtained from the difference between the micro and the

total pore volume (Table 1), varies considerably between the materials. The NLDFT pore-size distributions (PSD) highlight the substantial micropore volume (Supplementary Fig. 13), whereas the BJH analysis (Supplementary Fig. 14) confirms significant volumes of wide mesoporosity (varying between 2 and 20 nm for monoliths UiO-66_A-D). Small mesopores have previously been observed in UiO-66, on account of the enlargement of micropores due to missing linker/cluster defects[43]. Crucially, both the size and volume of the mesopores obtained in this study vary with drying conditions. The observed meso/macroporosity is not the result of crystalline defects within the MOF primary particles and instead corresponds to void space in-between them, enforced by their arrangement in the macrostructure. The synthetic origin of the tuneable mesoporosity thus lies in the variations to primary particle drying conditions which alter particle packing/densification. Overall, the mesopore volumes follow the trend $_{mono}$UiO-66_A > $_{mono}$UiO-66_B > $_{mono}$UiO-66_C > $_{mono}$UiO-66_D.

We analysed the adsorption behaviour of the samples using computational simulations of mixed micro-/mesoporous UiO-66 by artificially creating a mesopore between two purely

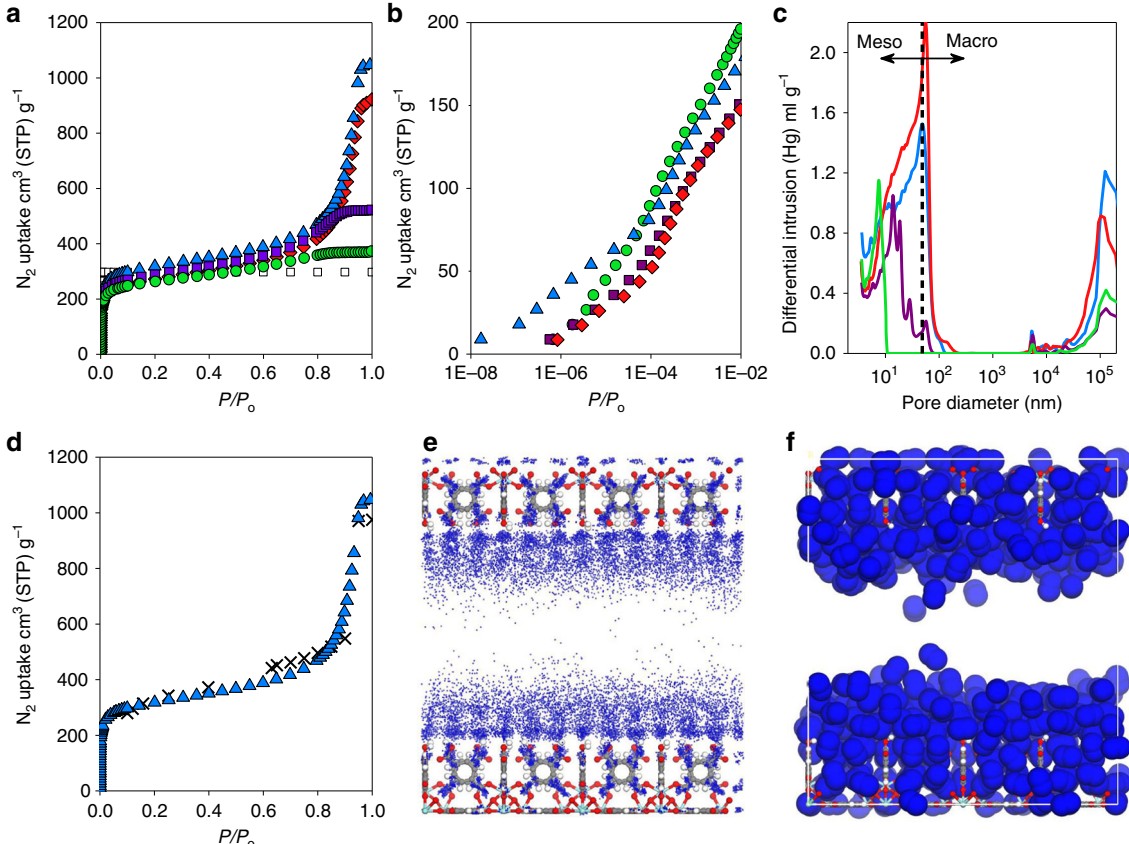

**Fig. 4** N$_2$ adsorption isotherms and Hg-porosimetry pore-size distributions of UiO-66 monoliths. **a** Linear and **b**, semi-log isotherm plots of gravimetric N$_2$ uptake at 77 K for monolithic UiO-66_A (blue triangles), UiO-66_B (red diamonds), UiO-66_C (purple squares) and UiO-66_D (green circles). The theoretical isotherm (white squares) was simulated from the crystal structure of UiO-66 (Supplementary Fig. 15). The experimental isotherms indicate extensive microporosity, which is comparable with the simulated isotherm, as well as variable mesoporosity, indicated by N$_2$ uptake at higher relative pressure. **c** Pore-size distributions (PSD) obtained by Hg-porosimetry showing the variations in meso- (2–50 nm diameter) and macro- (> 50 nm diameter) porosity amongst the materials. **d** Comparison of experimental $_{mono}$UiO-66_A (blue triangle) and simulated micro-/mesoporous UiO-66 (black cross), indicating the good match obtained when using an appropriate ratio of porosities (see the Methods section). **e, f** The density distribution and a snapshot, respectively, of simulated N$_2$ adsorption (navy blue spheres) at 0.7 $P/P_o$ (i.e., before capillary condensation — the micropores and the mesopore walls are completely filled by N$_2$, whereas the centre of the mesopore remains empty) between two UiO-66 crystals (Zr; blue, O; red, C; grey and H; white) separated by a 2.3 nm mesopore (representative of $_{mono}$UiO-66_D)

**Table 1 Textural and adsorption properties of monolithic UiO-66**

| | $S_{BET}$ | $S_{BET}$ (v) | $W_o$* | $V_{tot}$† | $V_{tot}$ (v) | $\rho_b$‡ | $CO_2$ 40 bar¥ | | $CH_4$ 65 bar¥ | | $CH_4$ 100 bar¥ | |
|---|---|---|---|---|---|---|---|---|---|---|---|---|
| | m$^2$ g$^{-1}$ | m$^2$ cm$^{-3}$ | cm$^3$ g$^{-1}$ | cm$^3$ g$^{-1}$ | cm$^3$ cm$^{-3}$ | g cm$^{-3}$ | g g$^{-1}$ | cm$^3$ cm$^{-3}$ | g g$^{-1}$ | cm$^3$ cm$^{-3}$ | g g$^{-1}$ | cm$^3$ cm$^{-3}$ |
| UiO-66_A | 1177 | 506 | 0.46 | 1.62 | 0.697 | 0.430 | 0.670 | 149 | 0.180 | 109 | 0.252 | 153 |
| UiO-66_B | 994 | 431 | 0.39 | 1.43 | 0.616 | 0.434 | 0.648 | 142 | 0.160 | 98 | 0.228 | 142 |
| UiO-66_C | 1065 | 903 | 0.42 | 0.81 | 0.687 | 0.848 | 0.635 | 265 | 0.170 | 202 | 0.243 | 289 |
| UiO-66_D | 982 | 1032 | 0.38 | 0.57 | 0.601 | 1.051 | 0.537 | 284 | 0.143 | 211 | 0.203 | 296 |
| Simulation | 1644 | 2033 | 0.49 | 0.49 | 0.602 | 1.237 | 0.375 | 236 | 0.103 | 178 | 0.116 | 200 |

BET area ($S_{BET}$), micropore volume ($W_o$), total pore volume ($V_{tot}$), bulk density ($\rho_b$) and gravimetric and volumetric $CO_2$ and $CH_4$ uptake measured for monolithic UiO-66_A-D and from simulated defect-free UiO-66
*Obtained at $P/P_o$ = 0.1; †obtained at $P/P_o$ = 0.98; ‡quantified using Hg-porosimetry; ¥the total absolute uptake

microporous UiO-66 layers (Supplementary Fig. 15). We were able to both match the condensation of N$_2$, i.e., the relative pressure at which N$_2$ uptake into the mesopores occurs and the characteristic step in N$_2$ uptake seen in Type IV isotherms (Fig. 4d; Supplementary Fig. 16). Although pore-shape is known to significantly influence gas adsorption behaviour[12], we found

that, by tuning the pore width in a model slit-shaped pore between 2.3 and 2.75 nm, we could predict computational isotherms that closely resembled those obtained experimentally. In this case, we found that a slit pore is representative of the wide mesoporous gap between two adjacent primary MOF particles— the origin of the mesoporosity in the current materials. Moreover,

by defining the ratio of micro-/mesoporosity, we achieved an excellent match of the whole experimental adsorption isotherms. The micro-/mesoporosity ratios obtained from the simulations were 0.45, 0.43, 0.71 and 0.85 for $_{mono}$UiO-66_A-D respectively. Calculation of the density distributions and snapshots during $N_2$ adsorption let us elucidate the most probable regions for adsorption and visualise how the pores are filled. Figure 4e and f show that, at 0.7 $P/P_o$ —i.e., before the onset of capillary condensation — the micropores and the mesopore walls are completely filled by $N_2$, whereas the centre of the mesopore remains empty.

To further study the monolith density and PSD in the meso- and macroporous range, we utilised Hg-porosimetry. Figure 4c shows that the experimental PSDs for the four samples match well those calculated by BJH (Supplementary Fig. 14). These data establish the importance of the materials showing both variable meso- and macroporosity. Although larger pore volumes are generally associated with enhanced adsorption capacities on a gravimetric basis, they also incur lower volumetric adsorption. When we quantified the bulk density ($\rho_b$) of the samples, we found that, as expected, those with greater meso- and macropore volumes showed lower densities (Table 1). Thus, $\rho_b$ in the monolithic MOFs varies according to UiO-66_D > UiO-66_C > UiO-66_B > UiO-66_A. Excitingly, $\rho_b$ of $_{mono}$UiO-66_D (1.05 g cm$^{-3}$) approaches the theoretical crystal density of UiO-66 (1.237 g cm$^{-3}$). We postulate that this relatively high density derives from the efficient packing of small primary particles within the monolithic macrostructure, with the extraneous interparticle space, which reduces $\rho_b$ in e.g., powdered materials/pellets, being minimised[44,45]. For example, Dhainut et al. previously reported the densification of UiO-66 powder at 18 MPa to obtain a UiO-66 pellet with 0.43 g cm$^{-3}$ density[46]. This pressure was selected as a compromise between maximising industrially significant physical properties, such as pellet density and mechanical strength, while minimising compressive loss of $S_{BET}$. Crucially, $_{mono}$UiO-66_D exhibits not only exceptional $\rho_b$ but also high $S_{BET}$ and relatively low but significant mesopore volume.

The simple modifications made to the original synthetic protocol (discussed above) account for the differences in physical properties between these materials. The near identical densities of highly mesoporous $_{mono}$UiO-66_A ($\rho_b = 0.430$ g cm$^{-3}$) and $_{mono}$UiO-66_B ($\rho_b = 0.434$ g cm$^{-3}$), dried at 200 and 30 °C, respectively, suggest that $\rho_b$ is not strongly dependent on drying temperature. In contrast, the high density of $_{mono}$UiO-66_C ($\rho_b = 0.834$ g cm$^{-3}$) demonstrates a significant reduction in interstitial space/mesoporosity, with the only synthetic difference being the washing solvent (i.e., DMF). The slow evaporation of this solvent during drying enables both primary particle densification and continued primary particle interaction (as supported by the FLIM data), controlling both the density and porosity of the monolith. Finally, at 1.05 g cm$^{-3}$, $_{mono}$UiO-66_D reveals the highest $\rho_b$. In this case, the extended centrifugation period applied prior to drying (30 °C) evidently facilitates better primary particle compaction and minimises meso-/macroporosity. These observations are especially significant when the high $S_{BET}$ area and microporosity of each monolith is considered. Pore collapse in materials obtained through traditional densification procedures (e.g., applied pressures) renders them unsuitable for physisorptative applications (e.g., dense gas storage)[16]. We have thus demonstrated that both the density and PSD of pure monolithic MOFs can be synthetically controlled without significant collapse/blocking of microporosity. The high thermal and mechanical stability of these robust and high density UiO-66 monoliths points strongly to their viability for commercial gas storage.

**Gas storage and selectivity.** We evaluated the gas adsorption performance of $_{mono}$UiO-66_A-D (Supplementary Figs. 17–20). Figure 5 shows the adsorption isotherms for $CH_4$ and $CO_2$; Table 1 compares the gas uptake of $CO_2$ (40 bar) and $CH_4$ (65 and 100 bar). Rapid, industrially viable equilibration was reached for both $CO_2$ and $CH_4$ (<270 s for all monoliths, comparable with $_{mono}$HKUST-1)[28], with no trend discernible related to monolith density or porosity (Supplementary Fig. 21). In addition, no significant differences in $CH_4$:$CO_2$ selectivity were observed between $_{mono}$UiO-66_A-D (Supplementary Figs. 22, 23), with each variant demonstrating high preferential uptake of $CO_2$.

Instead of the expected Type I isotherms, we observed surprising Type II isotherms. The presence of a quasi-linear performance in the high-pressure adsorption isotherms, compared with the traditional Type I isotherm in purely microporous samples (i.e., $_{mono}$HKUST-1, Fig. 5h, i), is of paramount importance from a technical point of view in vehicles under driving conditions—i.e., for a constant supply of gas–fuel upon a pressure change. The gravimetric uptake capacities recorded do not vary substantially across $_{mono}$UiO-66_A-D, reaching values of 0.252 g g$^{-1}$ (298 K, 100 bar) for $CH_4$ and of 0.670 g g$^{-1}$ (298 K, 40 bar) for $CO_2$. For comparison, Wu et al. reported $CH_4$ and $CO_2$ uptakes of 0.11 g g$^{-1}$ (300 K, 63 bar) and 0.38 g g$^{-1}$ (300 K, 30 bar), respectively. Notably, the exclusively microporous character of their powder UiO-66 meant that the porosity became saturated at relatively low pressures yielding plateaued Type I isotherms. In contrast, isotherms obtained for our mixed micro-/mesoporous $_{mono}$UiO-66 do not saturate at low pressures, demonstrating continued gas uptake even at the maximum pressures tested in this study. These promising Type II isotherms yield significant improvements in the total gas-storage capacity amongst UiO-66_A-D; 0.14–0.18 g g$^{-1}$ (298 K, 63 bar) for $CH_4$ and 0.43–0.54 g g$^{-1}$ (298 K, 30 bar) for $CO_2$, which we attribute to the narrow mesopore filling mechanism.

These results raise the question: Can the synthetic addition of mesoporosity be used to enhance gas adsorption capacity? While the presence of mesoporosity in the monoliths resulted in Type II isotherms with increased gas loadings at higher pressures, we observed little difference between the samples in terms of their gravimetric capacity (Fig. 5a, c). Yet the dramatic effect that these pores have on monolith density ($\rho_b$, Table 1) incurred significant changes to the volumetric adsorption capacity. Figure 5b, d and Supplementary Figs. 18, 20 show the volumetric adsorption isotherms for $CO_2$ and $CH_4$. First of all, each monolith displays an outstanding improvement in volumetric gas storage relative to a pressurised empty tank. Small differences in porosity and BET surface area between the samples have a minor influence on the total volumetric uptake, while the different densities of the monoliths cause outstanding changes. Volumetric gas uptake for both $CH_4$ and $CO_2$ follows the same trend: $_{mono}$UiO-66_D ≈ $_{mono}$UiO-66_C > > $_{mono}$UiO-66_B ≈ $_{mono}$UiO-66_A (Table 1). Remarkably, this is the trend we found for the micro-/mesoporosity ratio analysis during our molecular simulations described above, where $_{mono}$UiO-66_D had the largest relative amount of microporosity of the four experimental samples.

Overall, $_{mono}$UiO-66_D showed outstanding total gas uptakes for both $CH_4$ (211 and 296 cm$^3$ (STP) cm$^{-3}$ at 65 and 100 bar) and $CO_2$ (284 cm$^3$ (STP) cm$^{-3}$ at 40 bar) in a conformed material. Comparison with microporous UiO-66 (Table 1) shows that these results are significantly higher than the theoretical maximum. We attribute the performance of monolithic UiO-66 to the tuneable physical properties of these dense materials: high microporosity but weak $CH_4$ interaction facilitate moderate uptake at low pressures while mesopores permit Type II isotherms and enhanced gas condensation at higher pressures. Density distributions and snapshots of gas adsorption (Fig. 5e, f

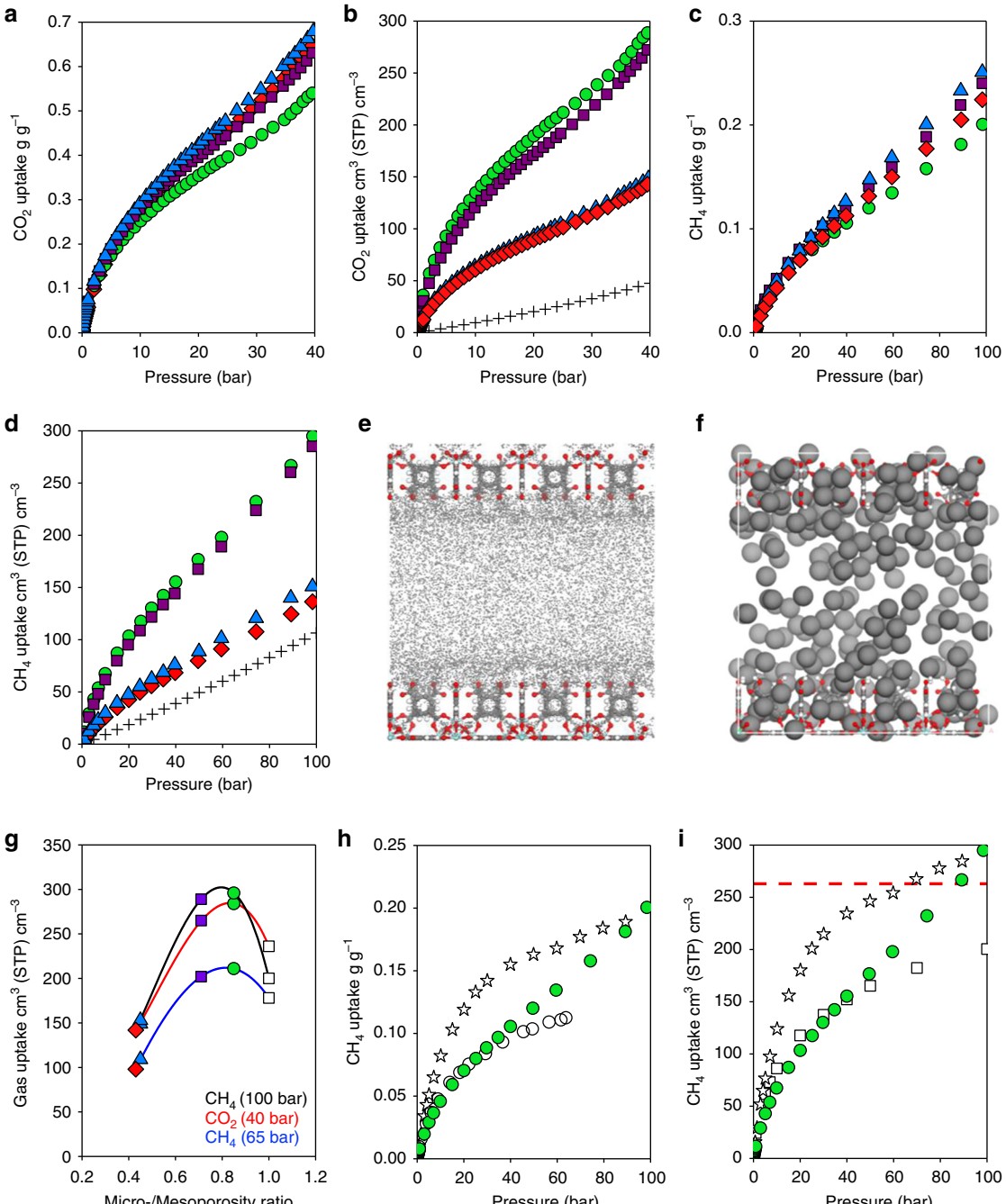

**Fig. 5** Experimental and computational $CO_2$ and $CH_4$ adsorption. **a** Gravimetric and **b** volumetric $CO_2$ absolute adsorption isotherms at 298 K.
**c** Gravimetric and **d** volumetric $CH_4$ absolute adsorption isotherms at 298 K. The data correspond to monoliths UiO-66_A (blue triangles), UiO-66_B (red diamonds), UiO-66_C (purple squares), UiO-66_D (green circles) and an empty tank (black crosses). **e** The density distribution and **f**, a snapshot of $CH_4$ (grey spheres) uptake into UiO-66 crystals (Zr; blue, O; red, C; grey and H; white) separated by a 2.3 nm mesopore (comparable with $_{mono}$UiO-66_D) at 80 bar. **g** Gas uptake amongst the different experimental and GCMC isotherms as a function of micro-/mesopore ratio, highlighting the ratio at which maximum uptake occurs. **h** Comparison of gravimetric absolute $CH_4$ adsorption isotherms for $_{mono}$UiO-66_D, UiO-66 powder (white circles)[47] and $_{mono}$HKUST-1 (white stars)[28]. **i** Comparison of experimental isotherms for absolute volumetric $CH_4$ uptake in $_{mono}$UiO-66_D (green circles) and $_{mono}$HKUST-1 (white stars)[28] to computationally simulated purely microporous/defect-free UiO-66 (white squares) at 298 K; the U.S. DOE volumetric $CH_4$ storage target of 263 cm$^3$ (STP) cm$^{-3}$ (65 bar) is indicated by the dashed red line

and Supplementary Fig. 25) demonstrate this phenomenon, showing gas condensation in optimised mesopores at high pressure, which thus leads to an overall enhancement in volumetric gas-uptake capacity relative to the purely microporous material. If mesoporosity is too extensive, such as in $_{mono}$UiO-66_A-B, the material's low density translates into low volumetric

gas storage. Figure 5g clearly demonstrates the existence of this maximum performance in terms of gas-uptake capacity, correlated with an optimal amount of mesoporosity. In $_{mono}$UiO-66_C-D, we achieved a finely tuned balance between surface area, porosity and density for optimised gas uptake, highlighting the power of molecular simulations to identify trends in this area.

When benchmarking our data against available literature, we first compared the $CH_4$ isotherms of $_{mono}$UiO-66_D with that of powdered UiO-66[47] (Fig. 5h). The isotherms of both UiO-66 materials show similar uptake up to ca. 30 bar, attributable to the filling of comparable microporosity in each material. However, above this pressure, $CH_4$ uptake by $_{mono}$UiO-66_D continues to increase, whereas that of powdered UiO-66 plateaus. $\rho_b$ for powdered UiO-66 was not reported, preventing comparison in terms of volumetric capacity. Using GCMC simulations of defect-free microporous UiO-66, we obtained a theoretical $CH_4$ isotherm that closely matched the results for UiO-66 powder (Supplementary Fig. 24). It is well reported that defects i.e., missing linkers, missing clusters and non-porous phases, are common amongst experimentally obtained UiO-66[47,48]. The similarity of these isotherms demonstrates that, at relatively high temperature, i.e., 300 K, the potential errors in the force field and pore volume estimation using a pristine crystalline UiO-66 with perfect activation on the one hand, versus a real UiO-66 with missing linkers and potential non-porous phases on the other hand, counter-balance. At the end of the day, crystalline defects, typical to UiO-66, incur only a minor influence on methane-adsorption capacity compared with the non-crystalline mesoporous defects in $_{mono}$UiO-66. Using the single-crystal density of the MOF, we found that the simulated volumetric $CH_4$ capacity of theoretical microporous UiO-66 again matches the volumetric isotherm for UiO-66_D up to only ca. 50 bar (Fig. 5i). The fact that mixed micro-/mesoporous $_{mono}$UiO-66_D closely matches both the theoretical and experimental UiO-66 isotherms at lower pressures, while exceeding both at higher pressures, supports the hypothesis that enhanced gas uptake occurs through condensation in the synthetically introduced mesoporous cavities.

As described above, one of the most important practical engineering parameters for an optimal NG adsorbent is the working capacity. This is the uptake at maximum storage pressure minus the uptake at release pressure (typically ca. 5 bar)—i.e., the real volume of accessible gas in the storage system[49]. At the DOE's target pressure of 65 bar, $_{mono}$UiO-66_D shows a working capacity of 172 cm$^3$ (STP) cm$^{-3}$ using real bulk density. This can be compared with the theoretical benchmark ca. 200 cm$^3$ (STP) cm$^{-3}$ working capacity of powdered UTSA-76a, which, under packing, would be expected to display a 25–50% reduction in volumetric capacity down to 100–150 cm$^3$ (STP) cm$^{-3}$. This means that, at 172 cm$^3$ (STP) cm$^{-3}$, densified UiO-66_D surpasses this benchmark significantly. Furthermore, by assuming a Langmuir fitting and using single-crystal density, the volumetric working capacity (5–100 bar) of UTSA-76a will be 236 cm$^3$ (STP) cm$^{-3}$. This theoretical maximum value will fall to 118–177 cm$^3$ (STP) cm$^{-3}$ in a standard pellet. Our monolith, UiO-66_D shows a volumetric working capacity (5–100 bar) of 261 cm$^3$ (STP) cm$^{-3}$.

When comparing with the 191 cm$^3$ (STP) cm$^{-3}$ (5–65 bar) working capacity of the chemically unstable $_{mono}$HKUST-1, this result is, however, 10% lower. Although 65 bar is considered an optimal storage pressure, being easily obtained by low-cost single-stage compressors, higher pressures are increasingly being considered both industrially viable and safe. For example, the Toyota Mirai fuel-cell vehicle utilises a hydrogen tank stored under 700 bar pressure[29]. In the current case, a maximum pressure of 100 bar, though above the DOE target, represents significantly milder storage conditions than the 250 bar required for CNG. Yet, even at these higher pressures, the $_{mono}$HKUST-1 working capacity increases only up to 235 cm$^3$ (STP) cm$^{-3}$ (5–100 bar). This small increment is commonly observed amongst Type I microporous MOFs (Fig. 1a) that saturate at low pressure; substantial increases in gas uptake are prevented even at increased pressures. Interestingly at ca. 90 bar, $_{mono}$UiO-66_D overtakes $_{mono}$HKUST-1 and reaches a volumetric working capacity of 261 cm$^3$ (STP) cm$^{-3}$ (5–100 bar).

This outstanding result is a consequence of the material's finely tuned physical properties i.e., the unique combination of weak $CH_4$ interaction at low pressure and its Type II isotherm character, which enhances gas uptake at high pressure. Due to its extrinsic mesoporosity, UiO-66_D does not saturate even at 100 bar, representing an 11% improvement in working capacity relative to benchmark densified $_{mono}$HKUST-1 (5–100 bar). Furthermore, this significant improvement in volumetric working $CH_4$ storage capacity has been achieved with a densified, air-stable monolith. The working capacity of 261 cm$^3$ (STP) cm$^{-3}$ (5–100 bar) for $_{mono}$UiO-66_D is, to the best of our knowledge, the highest recorded for a chemically stable, conformed MOF over this pressure range.

We selected Zr-MOF UiO-66 for this study due to its industrially valued physical/chemical properties. However, this is just one of an extensive range of promising Zr-MOFs with different pore sizes, surface areas and chemical properties, making the entire family of Zr-MOFs industrially interesting[50]. To demonstrate the generality of the reported synthetic procedure for $_{mono}$Zr-MOFs, we further synthesised and fully characterised both amine-functionalised $_{mono}$UiO-66-NH$_2$ [Zr$_6$O$_4$(OH)$_4$(2-amino-1,4-benzenedicarboxylate)$_6$] with a range of different PSDs (Supplementary Figs. 26–36) and $_{mono}$NU-1000 (Zr$_6$($\mu_3$–O)$_8$(1,3,6,8-tetrakis(p-benzoate)pyrene)$_2$, Supplementary Figs. 37–43).

## Discussion

We have reported the synthesis of high density, centimetre-scale monoliths of UiO-66 without the use of high pressures or chemical binders. The obtained monoliths show high thermal, mechanical and chemical stability, characteristic of industrially viable Zr-MOFs. Crucially, we have developed a synthetic procedure that enables modulation of the extrinsic pore-size distribution within the monolithic macrostructures. Via the controlled variation of synthetic parameters, we have demonstrated that precise volumes of meso- and macroporosity, external to the MOFs crystalline structure, can be built into the material. Through the subsequent optimisation of both PSD and density, we have achieved benchmark results for gas storage. We studied $CH_4$ and $CO_2$ uptake capacity in a selection of $_{mono}$UiO-66 samples with high microporosity and varied mesoporosity/density, demonstrating Type II isotherms, which yield improvements in working capacity compared with previous work on UiO-66[47]. We used GCMC simulations to demonstrate that the physical origin of this enhanced uptake capacity lies in substantial gas condensation in the mesopores at high pressure.

The use of UiO-66, which demonstrates a relatively poor interaction with $CH_4$ at low pressure, combined with optimisation of its PSD to include finely tuned volumes of narrow mesoporosity, resulted in a volumetric $CH_4$ working capacity of 172 cm$^3$ (STP) cm$^{-3}$ at 65 bar and 261 cm$^3$ (STP) cm$^{-3}$ at 100 bar, an 11% improvement over the benchmark of $_{mono}$HKUST-1. Although a higher, yet still industrially reasonable, maximum working pressure is required, this result represents a step further in the design of materials for maximised $CH_4$ working capacity, making it unique among air-stable MOFs. This result not only expands our understanding of the experimental capabilities of porous MOFs but has major implications for real-world applications of MOFs in the crucial field of environmental fuel distribution.

## Methods

**Synthesis of UiO-66 gel**. Benzene-1,4-dicarboxylic acid (1.20 g, 7.25 mmol) and zirconium(IV) oxychloride octahydrate (1.61 g, 5.0 mmol) were dissolved in N,N-dimethylformamide (30 ml, 99 %). Concentrated hydrochloric acid (1.5 ml, 37 %) and glacial acetic acid (2.0 ml) were added with vigorous stirring. The resulting solution was sealed in a 100 ml Pyrex Schott bottle and heated to 100 °C for 2 h. This yielded UiO-66 as a thick white gel.

**Synthesis of UiO-66_A-D**. N,N-dimethylformamide (50 ml) was added to the UiO-66 gel as synthesised above and vigorously mixed. The diluted UiO-66 suspension (7.5 ml per tube) was centrifuged (3 min, 5500 rpm) and the supernatant decanted. The gel was washed, centrifuged (5500 rpm) and dried to produce a range of monoliths (Supplementary Table 1). The obtained monolith was soaked in acetone (3 × 5 ml, 24 h) and methanol (3 × 5 ml, 24 h) and then dried at room temperature overnight. It was then activated by heating to 110 °C under vacuum for 8 h.

**Characterisation**. TEM was performing using an FEI Philips Tecnai 20 and FEI Osiris STEM operated in scanning mode (accelerating voltage of 200 keV). Primary MOF particles were prepared for analysis by diluting undried MOF gel in acetone and briefly sonicating before coating onto a copper grid. Dried monoliths were prepared for analysis by crushing with a spatula and pressing a copper grid into the resulting powder. Images were processed with Image J software. Scanning electron microscope images were collected using a TESCAN MIRA3 FEG-SEM and processed with Image J software. PXRD patterns were collected using a PANalytical Empyrean diffractometer with an X'celerator detector (Cu-Kα$_1$ source, λ = 1.5406 Å). Monolith powders were prepared for PXRD analysis by gently crushing with a pestle and mortar before being placed on a zero-background silicon wafer. Inductively coupled plasma-optical emission spectroscopy (ICP-OES) and C, H and N elemental analyses were performed using a Thermo Scientific iCAP 7400 ICP-OES analyser and an Exeter analytical CE 440 elemental analyser (975 °C), respectively. Thermogravimetric analysis was collected using a Mettler Toledo/SDTA851 thermobalance with an alumina crucible. Low pressure N$_2$ isotherms (adsorption and desorption) were collected using a Micromeritics 3Flex at 77 K. Prior to analysis, samples were degassed in a vacuum oven at 110 °C for 8 h. In situ degas (110 °C, 8 h) was further performed after sample loading into the instrument, ensuring the total evacuation of MOF pores. Nanoindentation was performed using an MTS Nanoindenter XP in an isolation cabinet. Monoliths were loaded into an epoxy resin and polished flat to a 0.25 μm finish. Continuous stiffness measurement mode was used to obtain Youngs' modulus, Hardness and Load as a function of penetration depth up to 2000 nm using a Berkovich diamond tip. Poisson's ratio was set to 0.18, in accordance with prior indentation work on UiO-66[33]. Nanoindents were analysed by AFM using a Dimension ICON pro with RTESPA tip in tapping mode. Data were analysed using the Nanoscope Analysis software version 1.9. Monolith density at atmospheric pressure was estimated via mercury porosimetry using a POREMASTER-60 GT porosimeter from QUANTA-CHROME INSTRUMENTS up to a final pressure of 2000 bar. Prior to analysis, samples were degassed overnight under vacuum at 110 °C and in situ prior to testing. High-pressure uptake of both CH$_4$ and CO$_2$ was studied using a homemade fully automated manometric equipment designed and constructed by the LMA group and now commercialised by Quantachrome Instruments (i-sorbHP). Prior to analysis, samples were degassed overnight (120 °C under vacuum) and then again in situ.

Fluorescence lifetime measurements were performed using a house-built laser scanning confocal microscope equipped with pulsed interleaved excitation and time-correlated single photon counting detection, as described elsewhere[51]. A pulsed laser diode at 405 nm wavelength (LDH-P-C-405) was used for excitation of the different $_{mono}$UiO-66 samples. For the measurements, $_{mono}$UiO-66 MOFs were gently ground with a spatula, and the resulting powder was suspended in water and vortexed for ~3 min. From the suspension, ~30 μL was added to an eight-well LabTek I slide (VWR) and the UiO-66 fragments allowed to sediment. The surface was imaged using a ×60, 1.27 numerical aperture water-immersion objective (Plan Apo IR ×60 WI, Nikon). Typical scans of 100 μm × 100 μm were performed at a resolution of 500 pixels × 500 pixels, resulting in a pixel size of 200 nm. Magnified regions of 30 μm × 30 μm were collected with a resolution of 500 pixels × 500 pixels or a pixel size of 60 nm. To ensure a good signal-to-noise ratio while minimising the influence of photon pile-up and other high-signal artifacts, the count rate was kept between 50 and 500 kHz. This was achieved by adjusting the power of the 405 nm wavelength laser power between 2 and 10 μW, as measured at the sample using a slide power metre (S170C-Thorlabs). Image acquisition times of 100–200 s ensured the detection of 200–1000 photons per pixel, after which the phasor analysis was applied[40]. To improve the FLIM analysis, the data were spatially smoothed with a 3 × 3 pixels sliding window. All analysis was performed using the software framework PAM[52].

**Computational details**. Grand canonical Monte Carlo (GCMC) simulations were employed to obtain adsorption isotherms for N$_2$, CO$_2$ and CH$_4$ at 77 K and 298 K. Simulations were based on a model that included Lennard–Jones (LJ) interactions for the guest–guest and guest–host interactions. The LJ potential parameters for the framework atoms were taken from the Universal Force Field (UFF)[53]. The interactions involving N$_2$, CO$_2$[54] and CH$_4$[55] were described by the TraPPE force field. An atomistic representation was used for the MOF, starting from its crystal structure. The simulation cells consisted of 8 (2 × 2 × 2) unit cells for microporous UiO-66, and a single unit cell for all micro-/mesoporous structures, with a LJ cut-off radius of 12.8 Å and no tail corrections. For CO$_2$, long-range electrostatic interactions were handled using the Ewald summation technique. Periodic boundary conditions were applied in all three dimensions. For each pressure point, GCMC simulations consisted of 50,000 Monte Carlo cycles to guarantee

equilibration, followed by 50,000 production cycles to calculate the ensemble averages. All simulations included insertion/deletion, translation and rotation (for N$_2$ and CO$_2$) with equal probabilities.

GCMC simulations were run on five models; one corresponding to defect-free microporous UiO-66 and four corresponding to the micro-/mesoporous materials ($_{mono}$UiO-66_A-D). To obtain micro-/mesoporous materials, the microporous UiO-66 supercell (of eight unit cells) was modified by artificially creating a gap between two purely microporous UiO-66 layers, increasing the simulation cell length. The appropriate mesoporous gap lengths to match the experimental N$_2$ isotherms were 2.75 nm (for $_{mono}$UiO-66_A and _B), 2.50 nm (for $_{mono}$UiO-66_C) and 2.30 nm (for $_{mono}$UiO-66_D).

The ideal adsorbed solution theory (IAST) can accurately predict multicomponent isotherms from pure-component isotherms[56]. We employed the Python package pyIAST[57] to estimate the CO$_2$:CH$_4$ IAST selectivities for $_{mono}$UiO-66_A-D at 298 K and at different pressures from experimental pure-component CO$_2$ and CH$_4$ adsorption isotherms by using the BET and the quadratic model fittings for CO$_2$ and CH$_4$ respectively (Supplementary Figs. 22, 23).

## Data availability

The experimental dataset generated and/or analysed during the current study are available from the corresponding authors on reasonable request.

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

## Acknowledgements

This project has received funding from the European Research Council (ERC) under the European Union's Horizon 2020 research and innovation programme (NanoMOFdeli), ERC-2016-COG 726380. D.F.-J. thanks the Royal Society for funding through a University Research Fellowship. B.M.C. thanks the Ernest Oppenheimer Fund (Cambridge). J.S.A. would like to acknowledge financial support from MINECO (MAT2016–80285-p), GV (PROMETEOII/2014/004) and H2020 (MSCA-RISE-2016/NanoMed Project). J.G.L. acknowledges GV (GRISOLIAP/2016/089) for a research contract. J.P.M. would like to thank CRUK and The Cambridge Cancer Centre. P.Z.M. is grateful for start-up funds from the University of Sheffield. D.C.L. acknowledges the financial support of the Deutsche Forschungsgemeinschaft (DFG) through the collaborative research centre SFB1032 (Project B3) and the support of the LMU via the Center for NanoScience Munich (CeNS) and the LMUinnovativ BioImaging Network (BIN).

## Author contributions

B.M.C., A.E.H.W. and D.F.-J. designed the research; B.M.C. performed the main materials synthesis and characterisation; B.M.C. and J.P.M. performed the supporting materials synthesis and characterisation; B.M.C. and D.V. carried out the $N_2$ gas adsorption under the supervision of D.F.-J.; Computational experiments were performed by M.A.-A. under the supervision of P.Z.M. and D.F.-J.; $CO_2$ and $CH_4$ isotherms were collected by J.G.-L. under the supervision of J.S.-A. N.A.D., D.C.L. and S.W. carried out the FLIM analysis. B.M.C., A.E.H.W. and D.F.-J. wrote the paper with input from the rest of the authors. All the authors contributed to the final version.

## Additional information

**Competing interests:** D.F.-J. has a financial interest in the start-up company Immaterial Labs, which is seeking to commercialise metal–organic frameworks. The remaining authors declare no competing interests.

