## [Peer Review File · Nature Communications]

REVIEWERS' COMMENTS:

Reviewer #1 (Remarks to the Author):

I have reviewed all documents, with a particular focus on the authors' replies to previous referee comments. I do acknowledge that the authors have diligently updated and improved the manuscript in response to the various criticisms brought forth. The replies are generally thoughtful and detailed and—although I disagree with some of the details due to personal opinion—I do not want to unnecessarily drag out the review process and I find the answers generally acceptable.

However, from my point of view the main criticism remains—one of novelty. Although the authors have made arguments that the specifics of the current study differ from the two previously published ones on monoHKUST-1 and monoZIF-8 by the same group, I still consider it as a fine-tuning of synthesis conditions and adaption to the case of UiO66 more so than a major breakthrough. The authors do make a case that the outcome of the current study is of more practical relevance for actual gas storage, and I do agree with that. At this point, the editor needs to decide whether the novelty of this manuscript is sufficient for publication in Nat. Commun. If so, I think the manuscript can be published in its current form.